# Self-Supervised Pre-Training of Spiking Neural Networks by Contrasting Events and Frames

**Raghav Singhal**
Forschungszentrum Jülich
RWTH Aachen

**Jan Finkbeiner**
Forschungszentrum Jülich
RWTH Aachen

**Emre Neftci**
Forschungszentrum Jülich
RWTH Aachen

## Abstract

Artificial Neural Network (ANN) pre-training, followed by fine-tuning, is an established procedure to solve real-world problems where labeled data is scarce. Our work aims to adapt this established procedure to the domain of event-based vision and Spiking Neural Networks (SNNs). Event-based sensors, inspired by the retina, capture visual scenes with low latency and high dynamic range, making them suitable for many real-world vision problems. SNNs, inspired by biological neural networks, when implemented on neuromorphic hardware, enable energy-efficient and low-latency processing, making them well-suited for fully event-based pipelines. However, the lack of sufficiently large labeled datasets hinders the pre-training of SNNs. Here, we leverage joint frame and event data to forego labeling. We achieve this using self-supervised contrastive learning, where an ANN and SNN pair are jointly trained to assimilate (contrast) (un)related frame-event stream pairs. We show that the pre-trained SNN model reaches higher accuracy on several downstream visual classification benchmarks. These results signify that pre-training large-scale SNNs using raw data output from event cameras is possible and paves the way toward foundation SNN models.

## 1 Introduction

Inspired by the brain's remarkable capabilities, neuromorphic systems strive to capture its key mechanisms for low-power, versatile, and fast information processing [1–3]. Similar to the brain, neuromorphic hardware aims to directly leverage the physics of the hardware, leading to a many-fold increase in efficiency, parallelism, and response speed compared to conventional computers. While the algorithms for training such networks have progressed dramatically in the last decade [4–8], their applications still lag behind their conventional ANN counterparts. A key reason for this is the combined lack of suitable datasets, benchmarks, and training hardware, which collectively prohibit the highly practical workflow used in deep learning.

Pre-training is the standard practice in deep learning, thanks to improved deployment speed, generalizability, and robustness. This enables a common backbone network to extract meaningful low-level features for the task at hand. Pre-training relies critically on the availability of large datasets which capture various factors, such as background, noise, lighting, and inherent object variations [9]. This practice has become even more common in recent self-supervised and foundational models, whose training from scratch is often beyond the computational resources of most practitioners [10, 11].

38th Conference on Neural Information Processing Systems (NeurIPS 2024).

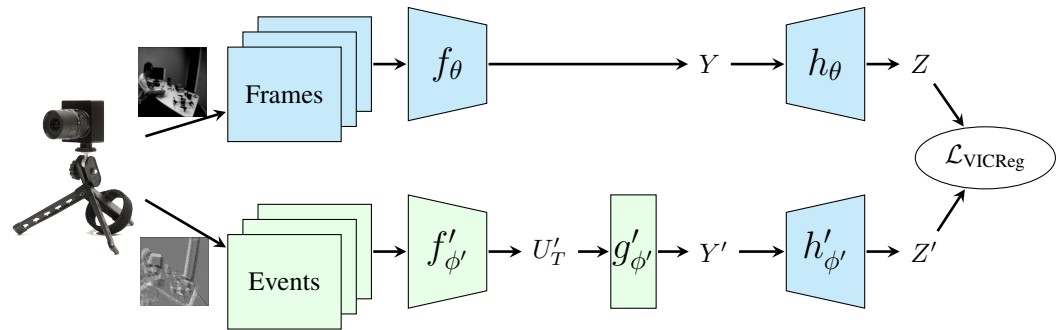

Figure 1: Given a batch of event streams and their corresponding frame-based images, we generate representations for images $Y$ by passing them through an encoder $f_\theta$. In parallel, event streams are encoded via $f_{\phi'}$ to produce a sequence of event representations $U'_T$, which are aggregated using an accumulator $g'_{\phi'}$, yielding representations $Y'$. These representations are fed into expanders $h_\theta$ and $h'_{\phi'}$ to produce embeddings $Z$ and $Z'$, which are jointly optimized using the VICReg loss. $f'_{\phi'}$ and $g'_{\phi'}$ (in green) are spiking in nature, while the other modules (in blue) are non-spiking.

Typical pre-training approaches are difficult to implement in neuromorphic computing due to the lack of labeled datasets on the scale of ImageNet and the higher training cost. While the latter is being addressed by more suitable hardware that emphasize locality and sparsity [12], the dataset labeling is a tedious process in videos and event streams. The lack of a neuromorphic SNN backbone is a critical obstacle to overcome before enabling their use in real-world applications. For example, in automotive traffic applications, the available event-based datasets are small compared to those of frame cameras [13]. This severely limits research in using novel vision sensors in real-world situations. A solid event-based vision backbone can mitigate this limitation by requiring a comparatively small fine-tuning dataset, which may be recorded in a well-controlled and consensual scenario.

Inspired by prior work in contrastive learning, we report here on a self-supervised algorithm to train a SNN on data recorded by neuromorphic vision sensors with simultaneous frame and event output. A majority of recent self-supervised techniques rely on data augmentations to achieve multiple views of the data [14, 15]. These different views are crucial to compute and minimize the similarity (contrast) losses [16–24]. Effective data augmentation techniques are critical in this context, however their application to event-based data has not been explored as extensively or rigorously as it has been for conventional image-based data. On the other hand, multimodal contrastive learning learns by associating pairs of data samples if they refer to the same object or concept. For example, CLIP [10] learns by associating a caption to an image if they are related while contrasting them if they are unrelated. Motivated by this approach, we show that it is possible to train an SNN using contrastive self-supervised learning by associating frame and event outputs to each other if they originate from the same visual sequence and disassociate them otherwise. This result is meaningful because very large datasets can be easily gathered using such cameras and pre-trained to provide a solid backbone for neuromorphic vision applications.

Our approach uses a joint architecture, as depicted in Figure 1, in which one branch is a conventional ANN and the other branch is an equivalent SNN. The ANN part embeds the frame output, while the SNN embeds the event output of the camera. The two branches are trained by minimizing the VICReg loss. The SNN follows Leaky Integrate and Fire (LIF) dynamics which are commonly used in software simulations and many digital and mixed-signal hardware emulations [25–29]. SNNs are compatible with modern ANN architectures such as Convnets [6, 30, 31] and transformers [32]. They can be trained using backpropagation through time using the well-established surrogate gradients approach [8]. We pre-train our model using a diverse corpus of frame-event pairs, synthesized from several open-source datasets, including DSEC [13], Brisbane-Event-VPR [33], DAVIS-DATA [34], MVSEC [35], and TUM-VIE [36]. Fine-tuning on downstream tasks, namely N-CARS [37] and CIFAR10-DVS [38], yields substantial performance improvements, demonstrating the effectiveness of our pre-training strategy. Our results demonstrate performance gains from self-supervised pre-training of SNNs, when fine-tuned on diverse downstream tasks.

## 2 Methods

**Core Methodology.** The overall architecture of our method is shown in Figure 1. We begin by processing a batch of event streams alongside their corresponding frame-based images. This data pairing, while originating from the same visual modality, can be viewed as multimodal since it captures different perspectives of the same sensory input. Image representations, $Y$, are generated by passing the images through an encoder, $f_\theta$. In parallel, event streams are processed by a SNN encoder, $f'_{\phi'}$, resulting in a sequence of spike-based representations, $U'_T$. To integrate these temporal spike representations, $U'_T$ is fed into an accumulator module, $g'_{\phi'}$, modeled as a Leaky Integrate layer with an infinite firing threshold. The final membrane potential, representing the cumulative spike activity across all time steps, forms the representation, $Y'$. Both sets of representations, $Y$ and $Y'$, are subsequently processed through their respective expander networks, $h_\theta$ and $h'_{\phi'}$, to produce the embeddings $Z$ and $Z'$.

These embeddings are optimized jointly using the VICReg loss function [39], ensuring robust alignment between the frame-based and event-based modalities (details about the loss can be found in [39]). VICReg is well-suited for this task as it independently regularizes each branch; it is thereby suitable for multimodal signal alignment tasks and helps to enhance the flexibility of joint-embedding self-supervised learning in such scenarios. During evaluation by fine-tuning on various downstream tasks, we only use the pre-trained SNN encoder, $f'_{\phi'}$, and discard the other modules.

**Pre-Training Dataset Processing.** The diversity and quality of data used during pre-training are critical for achieving robust model performance. We pre-train our model on a diverse corpus of frame-event pairs drawn from several event camera datasets, including DSEC [13], Brisbane-Event-VPR [33], DAVIS [34], MVSEC [35], and TUM-VIE [36]. Importantly, no ground truth labels are required, as our approach solely relies on the alignment between frames and event streams. Given the varying frame sampling rates across datasets, careful preprocessing is essential. For each dataset, we extract the last frame within each sampling period and aggregate the corresponding event stream into 10 time bins, ensuring consistency across the diverse sources. Through this process, we collected a total of 181,207 event-frame pairs across all datasets. A summary of the datasets is given in Table 1. DAVIS-DATA [34] includes 25 datasets featuring real-world event stream recordings

| Dataset | Image Frames | Sampling Period | Total Samples |
|---|---|---|---|
| DSEC [13] | RGB | 50 ms | 63,871 |
| Brisbane-Event-VPR [33] | RGB | 25 ms | 28,915 |
| DAVIS-DATA [34] | Grayscale | 46 ms | 25,825 |
| MVSEC [35] | Grayscale | 30 ms | 4,910 |
| TUM-VIE [36] | Grayscale | 50 ms | 57,686 |

Table 1: Overview of datasets used for pre-training. **Total Samples** denotes the total number of event-frame pairs we collected for each dataset.

alongside their corresponding greyscale frame-based images, encompassing scenarios such as office environments, urban settings, and simple objects and textures. The DSEC [13] and Brisbane-Event-VPR [33] datasets focus on event-based driving, capturing footage under a range of challenging lighting conditions. TUM-VIE [36] contains a diverse collection of handheld and head-mounted recordings in both indoor and outdoor contexts, including high dynamic range scenes. MVSEC [35] comprises recordings captured in various scenarios, including footage from a hexacopter, a car, and a motorcycle, all under different illumination levels and environmental conditions.

We expect that the diverse scenes represented in this collection of datasets will promote improved knowledge transfer during the subsequent fine-tuning process on other datasets.

**SNN Details.** The spiking neurons in our SNN follow the LIF model dynamics [40]. The dynamics are described by the following discrete-time equations:

$$S[t] = \Theta(H[t] - V_{th}) \tag{1}$$

$$H[t] = V[t-1] + \beta(X[t] - (V[t-1] - V_{reset})) \tag{2}$$

$$V[t] = H[t] \ (1 - S[t]) + V_{reset} \ S[t] \tag{3}$$

Here, $X[t]$ is the input current at time step $t$, $H[t]$ represents the membrane potential before spiking, and $V[t]$ is the membrane potential after a spike occurs. The neuron fires when $H[t]$ crosses the threshold $V_{th}$, which is determined by the Heaviside step function $\Theta(x)$. The spike output at time step $t$ is denoted by $S[t]$, while $V_{reset}$ is the reset potential after a spike. The constant $\beta$ governs the membrane decay. Since the step function is non-differentiable, we apply the surrogate gradient method [8], to approximate the derivative of the function, using the arctan surrogate function [41].

**Implementation Details.** We use PyTorch and SnnTorch [42] to simulate spiking neurons for training and Tonic [43] for dataset processing. The ANN and SNN encoders are the ResNet-18 model [44] and its adapted spiking variant [45], respectively. In the spiking ResNet-18 model, we replace all ReLU activations with leaky integrate-and-fire (LIF) modules and substitute max-pooling operations with average pooling layers. The membrane decay potential $\beta$ is set to $0.5$, and the reset potential $V_{reset}$ to $0$. Both the ANN and SNN encoder networks consist of $512$ output units. The expander networks comprise two-layer fully-connected models with batch normalization and ReLU activations, where each expander layer has $4096$ units, as larger expander dimension sizes have been shown to enhance performance [46]. Notably, neither the encoder nor the expander modules share weights. We resize each event and frame data sample to a spatial resolution of $224 \times 224$ pixels. Following the training protocols established by BYOL [47], Barlow Twins [46], and VICReg [39], we jointly train the models using the LARS optimizer [48], with a learning rate defined as $lr = base\_lr \times {}^{batch\_size}/256$, where $base\_lr$ is set to $0.2$ and the batch size is $128$. The learning rate follows a cosine decay schedule [49], starting from 0 with 10 warm-up epochs, ultimately reaching a final value of $0.002$. We use the same VICReg loss regularization coefficients as used in the original paper [39], and pre-train the models for 200 epochs.

**Evaluation on Downstream Tasks.** We evaluate our pre-trained SNN model on two downstream benchmark event-based datasets for visual classification: CIFAR10-DVS [38] and N-CARS [37]. We resize the spatial resolution of all samples to $128 \times 128$ pixels across both datasets. CIFAR10-DVS, derived from the original CIFAR10 dataset, consists of 10,000 DVS recordings. We use a 9:1 train-validation split, allocating 9,000 samples for training and 1,000 for validation. N-CARS, a real-world event-based dataset for binary object classification, comprises 15,422 training samples (7,940 cars, 7,482 background) and 8,607 test samples (4,396 cars, 4,211 background). For both datasets, we bin events into 10 time steps and set the membrane potential decay factor $\beta$ to $0.5$. We optimize our models using Adam [50] with an initial learning rate of $0.001$, following a cosine decay schedule [49], and a weight decay of $10^{-4}$. The results are reported as the average over 3 different runs.

## 3   Results and Conclusions

| Train Set % | w/o. PT | w. PT | ↑ Δ |
| --- | --- | --- | --- |
| 1 | 12.12 | 21.27 | **9.15** |
| 10 | 35.03 | 39.25 | **4.22** |
| 50 | 46.91 | 56.47 | **9.56** |
| 100 | 60.02 | 64.13 | **4.11** |

(a) CIFAR10-DVS

| Train Set % | w/o. PT | w. PT | ↑ Δ |
| --- | --- | --- | --- |
| 1 | 62.09 | 78.41 | **16.32** |
| 5 | 79.63 | 83.58 | **3.95** |
| 50 | 90.83 | 91.89 | **1.06** |
| 100 | 92.38 | 92.91 | **0.53** |

(b) N-CARS

Table 2: Transfer learning on downstream tasks. Test accuracy achieved through full fine-tuning (supervised training) of the spiking ResNet-18 model after training on varying percentages of the CIFAR10-DVS and N-CARS training sets, with a linear classifier appended. **w/o. PT** and **w. PT** refer to networks initialized without and with the proposed pre-training scheme, respectively. ↑ Δ represents the gain from utilizing the pre-training method over no pre-training.

Table 2 presents the performance of the spiking ResNet-18 model when fine-tuned on varying percentages of the CIFAR10-DVS and N-CARS training sets, with a linear classifier appended. Our proposed pre-training scheme consistently leads to a significant improvement in performance compared to the baseline without pre-training (fully supervised training). This improvement is especially notable when fine-tuning on smaller subsets of the training data, demonstrating the intended effectiveness of our method in low-data regimes. Our results show successful demonstration of self-supervised transfer learning in SNNs, with substantial gains observed on downstream datasets that were not part of the original training set. This highlights the model's capacity to transfer

knowledge and leverage representations learned during pre-training to enhance performance on unseen tasks. Additionally, the success of this approach opens new avenues for using SNN architectures in neuromorphic vision applications, laying a strong foundation for the development of versatile and generalizable SNN backbones. Our method also mitigates the challenges associated with limited exploration of data augmentation techniques for event-based data by simply enriching the event data with already existing frame-based images.

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

# A Appendix

## A.1 Related Work

Event-based cameras have garnered increasing interest due to their unique advantages over traditional frame-based cameras, including higher dynamic range and lower latency [51]. To process data from event-based cameras, one common approach is to slice event streams into frames, allowing standard architectures designed for image frames to be employed [52]. Alternatively, intensity images can be reconstructed from the event streams [53]. However, spiking neural networks (SNNs) offer a more natural fit for event-based data, particularly when considering the energy-efficient hardware available for SNNs [28, 54–57]. Despite this, efficiently training SNNs remains a significant challenge. A number of works have proposed methods to either distill or convert trained artificial neural networks (ANNs) into SNN models [58–60]. Advances in surrogate gradient techniques have made it possible to train SNNs in a manner that closely resembles standard deep learning methods [8, 61].

Large-scale model training, particularly for event-based data, increasingly relies on techniques that do not require manual annotation or labeled data. In the broader computer vision community, self-supervised learning (SSL) has gained prominence, with approaches like contrastive learning and joint-embedding architectures showing great promise [16–24]. These methods leverage data augmentations to create different views of the same image, but selecting the optimal combination of augmentations remains a challenge. Recent efforts have adapted SSL techniques to biologically-inspired settings, enabling some form of local learning in neural networks [62, 63], while others interpret the creation of different views in terms of causal and non-causal relationships in the world [64]. In addition, standard SSL techniques for image frames are now being adjusted for event-based data, with tailored augmentations for event streams [15, 65, 66]. There are also methods for converting video frames into event data for use in SSL [67–69].

In supervised learning, event-based data can be processed using standard ANNs as event frames [66, 70–72], hybrid SNN-ANN models [73, 74], or fully SNN-based architectures [15, 66]. To further enhance model performance, multimodal data can be incorporated instead of relying solely on vision-based data and random augmentations. Vision-language models (VLMs), such as CLIP [10], have shown strong performance when adapted to event-frames generated from event-based cameras [75]. Beyond text and vision, even different visual modalities, such as combining infrared and RGB data, have proven beneficial [76], with similar results observed when using RGB and event-based camera data together [77].

