# OpenReview forum: "Self-Supervised Pre-training of Spiking Neural Networks by Contrasting Events and Frames"
_NeurIPS.cc/2024/Workshop/UniReps — UniReps_

### Official Review · Reviewer_2Uyu · 2024-09-29
**A good paper with excessive citations**

**Rating:** 7
**Confidence:** 3

**Review:**

1. 77 cited articles for a 4 page workshop paper seems excessive. In line 45, citing 9 articles [16-24] seems unnecessary.
2. Are there other pre-training schemes in previous literature for SNNs?
3. How do you explain the test accuracy going up 4% with a 10% train set, then 9% with a 50% train set, when the test accuracy went up 9% with a 1% train set in the CIFAR10-DVS dataset? In the N-CARS dataset, the delta is decreasing consistently with the train set size increase.

---

### Official Review · Reviewer_Hqy1 · 2024-09-29
**Self-supervised contrastive learning for event-based images via alignment to conventional images**

**Rating:** 4
**Confidence:** 5

**Review:**

Spiking Neural Networks (SNNs) are a favorable paradigm in the domain of event-based computer vision. This paper proposes a self-supervised contrastive learning method for event-based vision to overcome the lack of labeled data. In particular, the proposed method has a dual pipeline with the conventional non-spiking model and a SNN that are optimized using the VICReg loss. The last stage of the dual architecture has a block to align spiking and non-spiking representations. Transfer learning experiments show that the SNN that have been pretrained using the proposed method performs better in the downstream tasks (classification) than a SNN without pretraining. Overall, the motivation of this paper is appealing but I believe, the proposed direction has a clear limitation. In particular, it requires to have paired data rather than trying to perform contrastive learning directly on the event-based data which is not available in all practical setups. Moreover, the "proxy" alignment approach can result in a complicated contrastive learning procedure with its own issues. Lastly, the proposed method has no brave academia contribution where the latter is the goal for workshop extended abstracts.

---

### Official Review · Reviewer_go23 · 2024-10-02
**Contrastive learning for Spiking**

**Rating:** 6
**Confidence:** 2

**Review:**

Though I am not an expert in Spiking Neural Networks (SNNs), the paper "Self-Supervised Pre-training of Spiking Neural Networks by Contrasting Events and Frames" presents a compelling and innovative approach. It addresses a key challenge in neuromorphic computing: the scarcity of labeled event-based datasets for pre-training SNNs. By leveraging contrastive learning between frame and event streams, the authors propose a self-supervised learning strategy that pairs an ANN with an SNN, enabling efficient pre-training without requiring labels. The method demonstrates significant improvements on downstream tasks like CIFAR10-DVS and N-CARS, especially when data is limited, indicating its effectiveness in low-data scenarios. The architecture and methodology are clearly outlined, and the results suggest potential for broader applications in neuromorphic vision. While the paper presents a solid contribution, further exploration into other datasets and architectures could help establish the approach's versatility and impact on the field. Overall, the paper appears to offer valuable insights into enhancing SNN performance through self-supervised learning.

I guess this could be an intersting contribution for UNIREPS, being the topic align the representations of ANN and SNN

---

### Decision · Program_Chairs · 2024-10-10

**Decision:**

Accept

**Comment:**

In light of the positive reviewers' feedback and relevancy of the submission, we are pleased to accept this paper for presentation at UniReps 2024. We kindly ask the authors to incorporate the reviewers' suggestions and feedback in the final camera-ready version of the manuscript.